# Predictors of maternal and newborn health service utilization across the continuum of care in Ethiopia: A multilevel analysis

Gizachew Tadele Tiruneh[1]*, Meaza Demissie[2], Alemayehu Worku[3], Yemane Berhane[2]

1 The Last Ten Kilometers (L10K) Project, JSI Research & Training Institute, Inc., Addis Ababa, Ethiopia,
2 Addis Continental Institute of Public Health, Addis Ababa, Ethiopia, 3 Addis Ababa University School of Public Health, Addis Ababa, Ethiopia

* gizt121@gmail.com

**Data Availability Statement:** The dataset used and analyzed during this study is included as supplementary information to this article (S2 Appendix).

## Abstract

### Background

The continuum of care for maternal and newborn health is a systematic approach for delivery of an integrated effective package of life-saving interventions throughout pregnancy, childbirth, and postpartum as well as across levels of service delivery to women and newborns. Nonetheless, in low-income countries, coverage of these interventions across the life cycle continuum is low. This study examined the predictors of utilization of maternal and newborn health care services along the continuum of care in Ethiopia.

### Methods

This was a cross-sectional population-based study. We measured maternal and newborn health care utilization practices among women who had live births in the last 12 months preceding the survey in Amhara, Oromia, SNNP, and Tigray regions of Ethiopia. We fitted multilevel random-effects logistic regression models to examine the predictors of the continuum of care accounting for the survey design, and individual, and contextual characteristics of the respondents.

### Results

Our analysis revealed that only one-fifth of women utilized maternal and newborn health services across the antepartum, intrapartum, and postpartum continuum; most women discontinued at the postpartum stage. Continued use of services varied significantly across wealth, model family, prenatal stay at maternity waiting homes, antenatal care in the first trimester, complete antenatal care service, and the administrative region at all antepartum, intrapartum, and postpartum stages. Moreover, family conversation during pregnancy [AOR: 2.12; 95% CI: 1.56–2.88], delivery by cesarean [AOR: 2.70; 95% CI: 1.82–4.02] and birth notified to health extension workers [AOR: 1.95; 95% CI: 1.56–2.43] were found to be predictors of the continuum of care at the postpartum stage.

**Funding:** JSI Research & Training Institute, Inc. has provided us support in the form of salaries for author [GT]. However, any of the funders did not have role in study design, data collection and analysis, decision to publish, or preparation of the manuscript.

**Competing interests:** The authors declare that they have no competing interests. The author [GT] has been working for JSI Research & Training Institute, Inc., a commercial company. We declared that this commercial affiliation does not alter our adherence to PLOS ONE policies on sharing data and materials.

**Abbreviations:** ANC, antenatal care; ANC4, four or more antenatal care visits; AOR, adjusted odds ratio; CI, confidence interval; CoC, continuum of care; HEP, Health Extension Program; HEW, Health Extension Worker; ICC, intra-class correlation coefficient; L10K, Last Ten Kilometers Project; MWH, maternity waiting home; PNC, postnatal care; SBA, skilled birth assistance; SNNP, Southern Nations, Nationalities, and Peoples' region.

## Conclusion

In Ethiopia, despite good access to antepartum care, compliance with continuity of care across the pathway decreased with significant inequitable distributions, the poorest segment of the population being at most disadvantage. The main modifiable program factors connected to the continued uptake of maternal health services include family conversation, pregnant women conference, complete antenatal care, antenatal care in the first trimester, and birth notification.

## Background

The continuum of maternal and newborn care is highlighted as a framework for delivering high-impact interventions across the pregnancy, childbirth, and postpartum life-course and the level of service delivery from community to the hospital to prevent maternal and newborn mortalities [1]. A recent systematic review also shows that the continuity of care from antepartum to postpartum periods may reduce the risk of combined neonatal, perinatal, and maternal mortality by 15% [2] and reduce neonatal and perinatal mortality risk by 21% and 16%, respectively [3].

In low-income countries, coverage of care is lowest during childbirth and postnatal period, and services are often fragmented further limiting the continuity of care [4–6]. The quality of antepartum and intrapartum care also influence women's healthcare-seeking decisions [7, 8]. Uptake of services can drastically decline from the antenatal to the postnatal period, along with the continuum of care (CoC), due to quality concerns [9, 10]. A major drop-out occurs early in the CoC; women who did not complete antenatal care (ANC) may not receive skilled delivery or postnatal care (PNC) services [11].

Uptake of maternal and newborn health services across the CoC is affected by factors such as those related to the women themselves and their households [12]. Individual-and household-level factors include marital status, maternal education, partner's education, wealth, women's autonomy in decision making, awareness regarding PNC, and neonate being ill [11, 13–18]. Additionally, availability of transportation, living in an area far from the health facility, content of antepartum care like urine tests, and delivery at a health facility were identified as predictors significantly associated with CoC [13].

Understanding factors that contribute to creating gaps across service use during antenatal, delivery, and postnatal care is imperative for the successful continuity of maternal and newborn health services and eventually for improvements in maternal and newborn health outcomes [19]. However, evidence on factors associated with adherence to the continuum of maternal and newborn health care is not widely available. Nationally, there is a critical knowledge gap regarding the characteristics of a continuum of maternal and newborn health services [20]. Why the utilization of maternal and newborn health services has not improved in Ethiopia despite the huge national efforts and investments made in the last decade is not well known. The main purpose of this study was to identify factors associated with the use of maternal and newborn health care along the pathway of CoC in Ethiopia.

## Methods

### Context

Administratively, Ethiopia is divided into 12 geographic regions where regions are divided into zones, which are internally divided into *woredas* (i.e., districts) and each woreda into the smallest administrative unit called *kebeles*.

In Ethiopia packages of maternal and newborn health interventions are delivered through home-based, community-based, and facility-based service delivery modalities. The country's health system has primary level care (encompassing a primary hospital, health centers, and health posts), secondary level care, and tertiary level care [21]. The primary health care provides preventive and promotive community and outreach services through the expansion of the Health Extension Program (HEP), the national flagship community-based health care delivery system, and the engagement of community volunteers [21, 22].

The country developed different strategies and programs to improve maternal survival which include strengthening and promoting skilled delivery through community mobilization [23]. The government has also provided ambulances to districts to mitigate transportation barriers, trained and deploy midwives and mid-level professionals, to improve access to, and utilization of maternal and newborn health services [22, 23].

In line with the national health agenda, the Last Ten Kilometers (L10K) project of JSI Research and Training Institute Inc. has supported the HEP to engage local communities to improve high-impact reproductive, maternal, newborn, and child health care behavior and practices in four of the most populous regions of the country (i.e., Amhara, Oromia, Southern Nations, Nationalities and Peoples [SNNP], and Tigray) since 2008. Between 2013 and 2017, the project scaled its platform activities in 115 woredas to engage local communities to identify and address barriers to access maternal and newborn health services particularly to identify pregnant women and ensure they received antenatal, intrapartum, and postpartum care. Besides, since 2014, L10K has implemented family conversation and birth notification strategies to promote birth preparedness, essential newborn care, and early postnatal care [24].

## Data source

The data used for this study were obtained from a cross-sectional population-based study representing the 115 rural woredas which were carried out by L10K Project from October-November 2017. The survey population included women of the reproductive age group (15–49 years) who had a live birth in the 12 months before the survey. The survey employed a two-stage stratified cluster sampling method stratified by the administrative region where kebeles were selected first as primary sampling units with the probability of selection being proportionate to its population size. This was undertaken to enumerate a representative sample of 2,724 women aged 15 to 49 years who had a live birth in the 12 months preceding the survey. The details of the design are described elsewhere [25].

The data were gathered through face-to-face interviews with mothers. During the interview, information about household and socio-demographic characteristics of mothers, awareness and access to health services, and experiences related to the women's use of maternal health services, was collected from women with children in their first year of life. The questionnaire (S1 Appendix) was translated into local languages (Amharic, Oromifa, and Tigrigna). Details of data collection processes are described elsewhere [26].

The adequacy of the sample size to address the study objective was assured considering 95% confidence level ($Z\alpha/2 = 1.96$), design effect ($D = 2$), and power of 80% for double population formula for comparative cross-sectional study design. Based on Anderson's health-seeking behavior model [27], a health service utilization model that provides a framework to systematically describe factors that influence individual decisions to use (or not use) health care services, researchers considered different exposure variables [5, 28] including lack of women's autonomy, 1–2 parity, no media exposure, no difficulty of distance to access medical care, no difficulty of transport arrangement to access medical care, no maternal education, and poorest wealth quintile as exposure and highest parity/5+, having autonomy, exposed to media, having

difficulty of distance and transport arrangement to access medical care, higher education, and richest wealth quintile as non-exposure. Adding a 10% non-response rate, the maximum sample size obtained by women's autonomy to healthcare decision-making was 2,501 for completed CoC at pregnancy [28].

## Measurement

The outcome variables of interest of the study were the uptake of the CoC at antepartum, intrapartum, and postpartum stages: 1) continuum of care at the antepartum stage is women who received four or more antenatal care (ANC4+) visits, 2) continuum of care at the intrapartum stage is those women who continued use of skilled birth attendance after receiving ANC4+ visits, and 3) continuum of care at the postpartum stage or complete continuum is those women who received PNC for the mothers and their newborns, within six weeks of their delivery (either in a facility or at home) after receiving both ANC4+ visits and delivered by skilled assistance. Description and measurement of variables are presented in Table 1 below.

For selecting predictor variables at individual and community levels, we adopted Anderson's behavioral model for healthcare use [27]. The individual-level variables include wealth status, maternal education, distance to the health facility, being a model family, participation in pregnancy conference, having family conversation, early ANC booking, complete ANC service, and use of MWHs for prenatal stay. Besides, infant's birth weight, mode of delivery and birth notification are included as predictor variables for CoC at the postpartum stage. The community variables considered in the study include region and area of residence (clusters/kebeles).

## Data analysis

Data were analyzed using Stata version 15. The characteristics of the sample respondents were described by a set of background characteristics. The difference in the characteristics of the respondents was examined using Pearson's chi-square statistics adjusted for cluster survey design effects.

Bivariate and multivariable mixed-effects logistic regression analyses were used to examine the predictors of the CoC accounting for cluster survey design, and the individual, and contextual characteristics of the respondents. We fitted three sequential random-effects logit regression models to examine the patterns of care-seeking and factors predicting the continuation of care. We fitted Model I among women receiving ANC4+ as the outcome (i.e., coded 1 for receiving ANC4+, otherwise 0); Model II among women who received ANC4+ to determine the factors associated with the continuity having skilled birth attendance (i.e., coded 1 for receiving ANC4+ and SBA, otherwise 0); and Model III fitted for women who received ANC4+ and SBA to identify factors associated with women returning for PNC visits or completion of the CoC (i.e., coded 1 for receiving ANC, SBA, and PNC, otherwise 0). The random-effects model accounts for the fact that people who live in the same area share similar characteristics and examine the proportion of variance explained by community-level factors (unobserved). We present the adjusted odds ratios and confidence intervals at the 95% level wherever applicable.

The global Wald's statistics, the likelihood ratio test of the cluster-level random effects, and sensitivity of the quadrature approximation were used to assess the goodness-of-fit of the models. Regression diagnostic Akaike Information Criterion (AIC) was used to determine the suitability of the model.

**Table 1. Description and measurement of variables, Ethiopia, 2017.**

| Variables | Descriptions | Measurements |
|---|---|---|
| Utilization of ANC services (ANC 4+ visits) | Having health facility visits at least 4 times for pregnancy check-ups by skilled attendants during pregnancy. It is defined as *continuum of care at ANC4+ (ANC4+ attainment)*. | Categorized into *at least 4 visits* and *less than 4 visits or none* |
| Skilled birth attendance (SBA) | It is defined as women who were assisted by a health professional (doctor, nurse, or midwife) during their last childbirth. | It is measured interview of women who were the primary person that assisted them with the delivery of their recent birth. |
| Postnatal care (PNC) | It is defined as women and their newborns who received postpartum care at the health facility or their home within six weeks of delivery. | It is measured interview of women whether pre-discharge care provided for them and their newborns after 24-hours of stay for whom delivered at the health facility and any postnatal check-up for mother's and newborn's health by a health care provider within 6 weeks of giving birth. The check-ups for the newborn and mother were separately inquired. |
| Continuum of care | In this study, the continuum of care is defined as the proportion of women who received maternal services at the pregnancy, delivery, and post-delivery stages. Accordingly, women who received all the following components were considered to have completed the continuum of care; | Continuum of care at delivery was obtained from ANC4+ and SBA variables; while a complete continuum of care was obtained from ANC4+, SBA, PNC variables |
| | 1. At least four ANC visits (ANC4+) by health service providers at a health facility or home, <br> 2. Delivery assisted by a health professional (i.e., doctor, nurse, midwife, or health officer/), and <br> 3. At least one PNC check-ups for mother within six weeks after delivery by health service providers at a health facility or home <br> 4. At least one PNC check-ups for mother within six weeks after delivery by health service providers at a health facility or home | The outcome for Model II is 1 for receiving antenatal care and skilled birth attendance, and 0 for receiving ANC but not SBA. After delivery, some women received PNC and some did not. Thus we fit Model III among women who received ANC and SBA to identify factors associated with completion of the CoC. The two categories of the outcome for Model III are 1 for receiving ANC, SBA, and PNC, and 0 for receiving ANC & SBA but not PNC. |
| Complete ANC | Complete ANC service was measured by the four content or items of care women received during ANC visits (i.e., blood pressure measured, weight taken, and blood and urine tested during last pregnancy). | Information on these items of ANC content was derived from the interviewing women whether they received these services as part of their ANC consultation during their last pregnancy. Based on the information, we created a complete ANC service if women received all four contents of care or otherwise no complete ANC service if women did not receive all four contents of care. |
| Distance from health facility | Approximate reporting walking distance from respondents home to the nearest health post or health center in minutes | The average distance was computed for each respondent and dichotomized as *< 30 minutes* and *> = 30 minutes* |
| Maternity waiting homes (MWHs) | These are residential facilities located near a hospital or a health center to accommodate women in their final weeks of pregnancy to bridge the geographical gap in obstetric care for women with poor access to facilities [29]. | Measured by asking whether respondents stayed at the MWHs during their last weeks of pregnancy or not |
| Family conversation | Family conversation is a forum conducted at the house of a pregnant woman with her family members and relatives who are expected to support her during pregnancy, labor, delivery, and the postpartum period. It creates an opportunity to discuss issues such as birth preparedness and essential newborn practices with all these family members together. | It is measured by asking women regarding their attendance of at least one family conversation at home during their last pregnancy. |
| Birth notification | It is a strategy introduced to promote early postnatal care. | It is measured through interviews of women whether they took measures to inform the Health Extension Workers (HEWs) about their childbirth immediately after delivery or not. |
| Pregnant women's conference | Pregnant women's conference is a pregnant women's group meeting for peer learning to seek maternal and newborn health care which is facilitated by health care providers. It creates an opportunity to discuss issues such as birth preparedness and essential newborn practices with all these family members together. | It is measured by asking women about their attendance at least one conference during their last pregnancy. |
| Model family | Model families are defined as those households who received training from HEWs, acquire the necessary knowledge, skills, and behavior in health practices and demonstrate practical changes in the use of health service programs and serve as models in their community [21]. | Measured by asking whether the respondent's family is currently recognized as a model family or not |

*(Continued)*

**Table 1.** (Continued)

| Variables | Descriptions | Measurements |
|---|---|---|
| Household wealth index | A wealth index score was constructed for each household with the principal component analysis of the household's possessions (electricity, watch, radio, television, mobile phone, telephone, refrigerator, table, chair, bed, electric stove, and kerosene lamp), and household characteristics (type of latrine, water source, floor, and wall material). Subsequently, households were ranked according to wealth score and then divided into five quintiles using the Principal Component Analysis method [30]. | Household assets ownership was assessed and the wealth index was computed by using principal component analysis. The wealth status was categorized into five groups and ranked from the poorest to the wealthiest quintile. |

## Ethics approval and consent to participate

For this study, we obtained permission to use the data from the JSI, and ethics approval was obtained from the Research and Ethics Committee of the Department of Health Studies of the University of Gondar (reference number V/P/RCS/05/2505/2019; dated 25 August 2019). The original study was ethically approved by the Ethical Review Boards of Amhara, Oromia, SNNP, and Tigray Regional Health Bureaus, and JSI. Verbal consent from respondents was sought and documented by interviewers before interviewing. Voluntary participation was ensured during interviews [25].

## Results

### Characteristics of the study participants

A total of 2,724 women with live births in the 12 months preceding the survey were included in this analysis. The socio-demographic characteristics of respondents are described elsewhere [26].

### Utilization of maternal and newborn health services

Most women (n = 2,481; 91.2% [95% CI: 89.1–93.0]) received at least one ANC during their recent pregnancy. However, only about a quarter of them (n = 698; 26.3% [95% CI: 23.9–28.9]) started ANC in their first trimester of pregnancy. A little more than half n = 1,387; 56.3% [95% CI: 54.5–58.2]) of them received complete ANC services (i.e., weight taken, blood pressure measured, blood test, and urine test conducted). About two-thirds (n = 1,816; 67.4% [95% CI: 62.8–71.6]) of them were attended by a skilled health provider (doctor, nurse, or mid-wife) during delivery at their last birth and about 5.6% of these were delivered through cesarean section. However, about 36.8% (n = 1,002; [95% CI: 33.3–40.4]) of women received PNC service within six weeks after delivery following their last childbirth; 29.4% of them received PNC within 48 hours; and 34.1% within 7 days.

### Continuum of maternal and newborn health care

As presented in Fig 1 of the error bar, a little higher than half (52.3%) of the women continued receiving ANC4+, 42.4% continued for skilled birth attendance, and about one-fifth (21.5%) of women received all three packages of services during pregnancy, childbirth, and postpartum periods. However, one-fifth of them did not receive any maternal and newborn health services. Also, around 8.4% of women receive at least one ANC from skilled providers but did not receive the other two services. About 12.4% of them received skilled birth assistance during childbirth without receiving pregnancy care. Similarly, few women received PNC (1.6%)

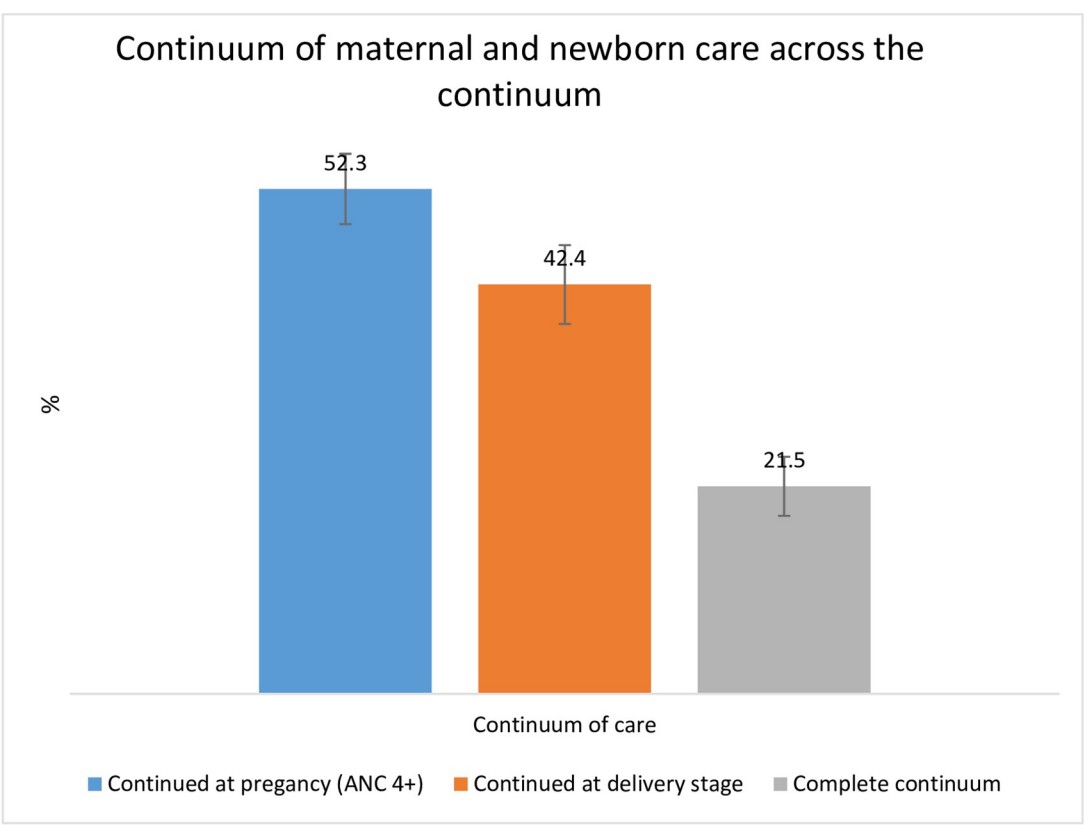

**Fig 1. Utilization of skilled maternal and newborn care across the continuum.**

without having antepartum and intrapartum care About 10% and 21% of women discontinued the CoC at the delivery and postnatal stage, respectively (Table 2).

The characteristics of women who completed different stages of care are shown in Tables 3 and 5. Of those women who resided within 30 minutes walking distance from the facility, 713 (55.3%) had four and above ANC visits, 584 (45.7%) continued to skilled delivery, 312 (24.4%) completed CoC at the postnatal stage. Besides, the distribution of the utilization of the package of maternal and newborn health services across the continuum significantly varied by region, educational status of the mother, and wealth quintile. Tigray and Oromia regions appeared to be better performing than Amhara region. Those mothers with better education residing near

**Table 2. Achievement of the continuum of maternal and newborn health care in Ethiopia, 2017.**

| Antepartum (ANC4+) | Intrapartum (ANC4+ & SBA) | Postpartum (ANC4+, SBA, & PNC) | Number (n) | Percent |
|---|---|---|---|---|
| Yes | Yes | Yes | 564 | 21.47 |
| Yes | Yes | No | 550 | 20.95 |
| Yes | No | Yes | 40 | 1.45 |
| Yes | No | No | 221 | 8.43 |
| No | Yes | Yes | 327 | 12.46 |
| No | Yes | No | 325 | 12.38 |
| No | No | Yes | 42 | 1.60 |
| No | No | No | 556 | 21.18 |
| **Total** | | | 2,626 | 100.00 |

**Table 3. Distribution of maternal health services across the CoC in Ethiopia, by sociodemographic characteristics, 2017.**

| Characteristics | Continued at antenatal care (received ANC 4) | | Continued at delivery stage | | Complete continuum (continued at postpartum) | |
|---|---|---|---|---|---|---|
| | n (%) | 95% CI | n (%) | 95% CI | n (%) | 95% CI |
| *Maternal age* | | | | | | |
| <20 | 83 (52.6) | 42.2–62.7 | 73 (46.6) | 36.6–56.8 | 32 (20.8) | 13.8–30.2 |
| 20–34 | 1,077 (52.6) | 48.9–56.3 | 859 (42.4) | 38.3–46.7 | 447 (22.1) | 19.0–25.4 |
| 35–49 | 227 (50.6) | 43.7–57.4 | 183 (41.0) | 34.2–48.2 | 85 (19.1) | 14.5–24.6 |
| *Education* | | | | | | |
| No education | 721 (47.5) | 43.4–51.7 | 538 (35.8) | 31.5–40.3 | 249 (16.6) | 13.7–19.9 |
| Primary | 310 (53.5) | 48.0–58.9 | 252 (43.8) | 37.7–50.0 | 141 (24.5) | 19.7–30.2 |
| Higher | 355 (64.1)* | 58.5–69.3 | 324 (59.2)* | 53.0–65.0 | 173 (31.7)* | 26.7–37.1 |
| *Religion* | | | | | | |
| Orthodox | 791 (50.2) | 45.7–54.8 | 464 (41.2) | 36.3–46.3 | 302 (19.3) | 16.0–23.2 |
| Protestant | 284 (56.7) | 50.2–62.9 | 223 (45.1) | 37.7–52.7 | 148 (29.8) | 23.1–37.7 |
| Muslims | 301 (54.1) | 45.5–62.5 | 242 (44.0) | 34.6–53.9 | 111 (20.2) | 15.3–26.1 |
| Traditional/others | 11 (53.9) | 36.2–70.7 | 5 (24.8) | 11.2–46.4 | 3 (16.7)** | 7.2–34.0 |
| *Wealth quintile* | | | | | | |
| Poorest | 224 (41.7) | 35.0–48.8 | 166 (31.0) | 25.0–37.6 | 68 (12.7) | 9.0–17.7 |
| Poorer | 265 (50.0) | 44.1–55.9 | 214 (40.7) | 34.4–47.5 | 105 (19.9) | 15.4–25.3 |
| Middle | 265 (50.1) | 44.1–56.1 | 210 (40.0) | 34.2–46.1 | 104 (19.8) | 15.4–25.0 |
| Richer | 312 (58.0) | 53.0–62.8 | 244 (46.4) | 40.5–52.5 | 138 (26.2) | 21.4–31.7 |
| Richest | 321 (61.8)* | 55.3–67.9 | 280 (54.5)* | 47.9–60.9 | 150 (29.1)* | 24.2–34.6 |
| *Distance to the nearest health facility* | | | | | | |
| <30 min | 713 (55.3) | 50.6–59.8 | 584 (45.7) | 40.6–50.8 | 312 (24.4) | 20.9–28.4 |
| >=30 min | 674 (49.5)** | 45.2–53.7 | 530 (39.3)** | 34.8–44.0 | 252 (18.7)* | 15.4–22.4 |
| *Region* | | | | | | |
| Amhara | 379 (42.3) | 36.5–48.7 | 285 (32.1) | 26.1–38.7 | 114 (12.9) | 9.0–18.0 |
| Oromia | 314 (45.2) | 39.0–51.6 | 229 (33.5) | 27.0–40.8 | 144 (21.0) | 15.8–27.5 |
| SNNP | 433 (64.5) | 58.8–69.9 | 368 (55.5) | 48.1–62.7 | 179 (27.0) | 21.5–33.3 |
| Tigray | 261 (66.1)* | 58.2–73.1 | 232 (59.2)* | 50.7–67.2 | 127 (32.4)* | 25.9–39.5 |
| **Total** | 2,653 (52.3) | 48.7–55.9 | 2,626 (42.4) | 38.5–46.5 | 2,626 (21.5) | 18.6–24.6 |

*P-value <0.001;

** P-value <0.05.

health facilities and those with better wealth appeared to better use the services across the continuum (Table 3).

As presented in Table 4, the distribution of maternal health services across the CoC also varied by health-service-related and obstetric characteristics. The CoC at antepartum, intrapartum, and postpartum stages significantly differed among women who stayed at MWHs, who were recognized as model family, who attended at least 3 pregnancy conferences during last pregnancy, those who attended at least one family conversation during pregnancy, received ANC in the first trimester, and received complete ANC, as compared to those who did not. While those women who delivered through cesarean section and their birth was notified to HEWs received complete CoC significantly better than their counterparts (Table 4).

## Predictors of the continuum of care

Except for maternal education, other covariates remain statistically significant in the multivariable analysis. As presented in Model I, Table 5, the use of ANC4+ visits at the stage of

**Table 4. Distribution of maternal health services across the CoC in Ethiopia, by health-service related and obstetric characteristics, 2017.**

| Characteristics | Continued at antenatal care (received ANC 4) | | Continued at delivery stage | | Complete continuum (continued at postpartum) | |
|---|---|---|---|---|---|---|
| | n (%) | 95% CI | n (%) | 95% CI | n (%) | 95% CI |
| *Prenatal stay at MWHs* | | | | | | |
| No | 1,075 (49.6) | 45.8–53.4 | 803 (37.5) | 33.3–41.9 | 394 (18.4) | 15.5–21.6 |
| Yes | 312 (64.2)* | 57.7–70.3 | 311 (64.2)* | 57.7–70.2 | 170 (35.2)* | 29.6–41.2 |
| *Family currently recognized as a model family* | | | | | | |
| No | 848 (47.1) | 43.2–51.0 | 648 (36.5) | 32.3–40.8 | 276 (15.5) | 13.1–18.3 |
| Yes | 539 (63.3)* | 58.3–68.0 | 466 (54.9)* | 49.4–60.4 | 288 (34.0)* | 29.2–39.1 |
| *Attended at least 3 pregnancy conference during last pregnancy* | | | | | | |
| No | 1,167 (50.1) | 46.4–53.8 | 929 (40.4) | 36.4–44.4 | 460 (20.0) | 17.3–23.0 |
| Yes | 220 (67.6)* | 60.0–74.4 | 185 (56.9)* | 47.4–66.0 | 104 (32.0)* | 24.2–40.9 |
| *At least one family conversation that took place at home during pregnancy* | | | | | | |
| No | 1,228 (50.9) | 47.2–54.5 | 970 (40.6) | 36.6–44.7 | 452 (18.9) | 16.3–21.8 |
| Yes | 159 (66.5)* | 57.7–74.3 | 144 (61.2)* | 52.4–69.4 | 112 (47.4)* | 38.1–57.0 |
| *ANC in the first trimester* | | | | | | |
| No | 912 (46.7) | 42.8–50.5 | 724 (37.5) | 33.4–41.7 | 364 (18.8) | 15.9–22.2 |
| Yes | 475 (68.0)* | 62.3–73.3 | 390 (56.2)* | 50.2–62.1 | 200 (28.8)* | 24.4–33.6 |
| *Complete ANC* | | | | | | |
| No | 494 (42.6) | 38.0–47.3 | 348 (30.2) | 25.9–34.9 | 171 (14.9) | 12.0–18.3 |
| Yes | 893 (59.9)* | 55.8–63.8 | 766 (51.9)* | 47.5–56.3 | 393 (26.6)* | 22.9–30.6 |
| *Mode of delivery* | | | | | | |
| Vaginal delivery | 1,282 (51.0) | 47.3–54.7 | 1,009 (40.6) | 36.5–44.8 | 503 (20.2) | 17.4–23.4 |
| Cesarean section delivery | 105 (75.5)* | 66.0–83.0 | 104 (75.5)* | 66.0–83.0 | 61 (44.2)* | 35.6–53.3 |
| *Birth notification* | | | | | | |
| No | 712 (45.9) | 41.4–50.4 | 537 (35.0) | 30.4–39.8 | 213 (13.9) | 11.3–16.9 |
| Yes | 675 (61.3)* | 57.0–65.4 | 577 (52.9)* | 48.2–57.5 | 351 (32.2)* | 27.8–36.8 |
| **Total** | 2,653 (52.3) | 48.7–55.9 | 2,626 (42.4) | 38.5–46.5 | 2,626 (21.5) | 18.6–24.6 |

*<0.001;

**<0.05.

pregnancy is significantly associated with the administrative region, wealth quintile, staying at MWHs, being model family, attendance at pregnancy conferences, and booked for ANC in the first trimester. Those women in the richest wealth quintile are more likely to complete the pathways to maternal and newborn health care. The odds of continued use of antepartum care among women who started ANC in the first trimester of pregnancy are about two times higher [adjusted odds ratio (AOR): 2.76; 95% confidence interval (CI): 2.24–3.41] than those who booked late.

Model II presents the predictors of continuation of care from pregnancy to delivery among women who received ANC 4+. All variables remain significant. Besides, higher maternal education is significantly associated with the CoC at the delivery stage [AOR: 1.32; 95% CI: 1.03–1.69]; those women who attended at least one family conversation during pregnancy had 43% higher odds of complete antepartum and intrapartum skilled care than those counterparts [AOR: 1.43; 95% CI: 1.04–1.96]; and those women whose homes were within less than 30 minutes walking distance to the health facility, had 30% more odds of continually utilizing package of services at delivery stage [AOR: 1.23; 95% CI: 1.02–1.50].

**Table 5. Factors associated with the CoC at antenatal, delivery, and postnatal care in Ethiopia, 2017.**

| Characteristics | Model 1 (ANC stage) | | | Model II (ANC4 & skilled delivery) | | | Model III (CoC) | | |
|---|---|---|---|---|---|---|---|---|---|
| | # of respondents | AOR (95% CI) | p-value | # of respondents | AOR (95% CI) | p-value | # of respondents | AOR (95% CI) | p-value |
| *Maternal education* | | | | | | | | | |
| No education | 1,518 | 1.00 | | 1,502 | 1.00 | | 1,502 | 1.00 | |
| Primary | 581 | 0.98 (0.79–1.23) | 0.893 | 576 | 1.00 (0.79–1.26) | 0.990 | 576 | 1.05 (0.81–1.36) | 0.708 |
| Higher | 555 | 1.07 (0.84–1.36) | 0.594 | 548 | 1.32 (1.03–1.69) | 0.030 | 548 | 1.20 (0.92–1.56) | 0.184 |
| *Wealth quintile* | | | | | | | | | |
| Poorest | 537 | 1.00 | | | 1.00 | | | 1.00 | |
| Poorer | 529 | 1.70 (1.28–2.25) | <0.001 | 535 | 1.80 (1.33–2.42) | <0.001 | 535 | 1.66 (1.17–2.34) | 0.001 |
| Middle | 529 | 1.59 (1.19–2.12) | 0.002 | 526 | 1.55 (1.14–2.10) | 0.005 | 526 | 1.42 (1.00–2.03) | 0.051 |
| Richer | 538 | 2.57 (1.89–3.51) | <0.001 | 525 | 2.41 (1.74–3.34) | <0.001 | 525 | 2.36 (1.64–3.39) | <0.001 |
| Richest | 520 | 2.61 (1.90–3.60) | <0.001 | 525 | 2.99 (2.14–4.18) | <0.001 | 525 | 2.03 (1.40–2.94) | <0.001 |
| *Distance to the health facility* | | | | 515 | | | | | |
| <30 min | 1,291 | 1.11 (0.93–1.34) | 0.254 | 1,278 | 1.23 (1.02–1.50) | 0.034 | 1,348 | 1.33 (1.07–1.64) | 0.009 |
| > = 30 min | 1,362 | 1.00 | | 1348 | 1.00 | | 1,278 | 1.00 | |
| *Family recognized as a model family* | | | | | | | | | |
| Yes | 852 | 1.57 (1.29–1.90) | <0.001 | 848 | 1.44 (1.18–1.78) | <0.001 | 848 | 1.59 (1.28–1.97) | <0.001 |
| No | 1,801 | 1.00 | | 1,778 | 1.00 | | 1,778 | 1.00 | |
| *Prenatal stay at MWHs* | | | | | | | | | |
| Yes | 486 | 1.38 (1.08–1.76) | 0.009 | 484 | 2.34 (1.82–3.00) | <0.001 | 484 | 1.97 (1.53–2.54) | <0.001 |
| No | 2,167 | 1.00 | | 2,142 | 1.00 | | 2,142 | 1.00 | |
| *Attended at least 3 pregnancy conferences during last pregnancy* | | | | | | | | | |
| Yes | 326 | 1.96 (1.46–2.62) | <0.001 | 325 | 1.77 (1.33–2.37) | <0.001 | 325 | 1.28 (0.95–1.71) | 0.105 |
| No | 2,327 | 1.00 | | 2,301 | 1.00 | | 2,301 | 1.00 | |
| *Attended at least 1 family conversation during pregnancy* | | | | | | | | | |
| Yes | 238 | 1.33 (0.97–1.82) | 0.077 | 236 | 1.43 (1.04–1.96) | 0.027 | 236 | 2.12 (1.56–2.88) | <0.001 |
| No | 2,415 | 1.00 | | 2,390 | 1.00 | | 2,390 | 1.00 | |
| *Antenatal care utilization in the first trimester* | | | | | | | | | |
| Yes | 698 | 2.76 (2.24–3.41) | <0.001 | 694 | 2.03 (1.65–2.49) | <0.001 | 694 | 1.46 (1.17–1.81) | 0.001 |
| No | 1,955 | 1.00 | | 1,932 | 1.00 | | 1,932 | 1.00 | |
| *Complete ANC* | | | | | | | | | |
| Yes | 1,491 | | | 1,478 | 2.13 (1.74–2.61) | <0.001 | 1,478 | 1.80 (1.43–2.26) | <0.001 |
| No | 1,162 | | | 1,150 | 1.00 | | 1,150 | 1.00 | |
| *Mode of delivery* | | | | | | | | | |
| Vaginal delivery | | | | | | | 2,488 | 1.00 | |
| Cesarean section delivery | | | | | | | 138 | 2.70 (1.82–4.02) | <0.001 |
| *Birth notification* | | | | | | | | | |
| Yes | | | | | | | | 1.95 (1.56–2.43) | <0.001 |
| No | | | | | | | | 1.00 | |
| *Community characteristics* | | | | | | | | | |
| *Region* | | | | | | | | | |
| Amhara | 893 | 1.00 | | 888 | | | 888 | 1.00 | |
| Oromia | 694 | 1.32 (0.94–1.86) | 0.107 | 682 | 1.49 (1.02–2.16) | 0.037 | 682 | 2.78 (1.86–4.16) | <0.001 |
| SNNP | 671 | 2.94 (2.09–4.13) | <0.001 | 663 | 3.30 (2.29–4.74) | <0.001 | 663 | 2.73 (1.87–3.99) | <0.001 |
| Tigray | 395 | 2.82 (1.97–4.04) | <0.001 | 393 | 3.36 (2.30–4.92) | <0.001 | 392 | 2.85 (1.91–4.24) | <0.001 |
| Community-level intercepts (SE) | | 0.22 (0.04) | | | 0.07 (0.01) | | | 0.02 (0.004) | |
| *Random effects* | | | | | | | | | |
| Community-level variance (SE) | | 0.39 (0.09) | | | 0.46 (0.10) | | | 0.34 (0.10) | |
| Log-likelihood ratio test | | 48.23* | | | 52.87* | | | 24.32* | |
| *Model fit statistics* | | | | | | | | | |
| ICC (SE) | | 0.11 (0.02) | | | 0.12 (0.02) | | | 0.09 (0.02) | |
| AIC | | 3212.72 | | | 3031.01 | | | 2539.54 | |

In Model III, administrative region, wealth quintile, being model family, having booked ANC in the first trimester, and having received complete ANC remained significantly associated with complete CoC at the postnatal stage among women who received ANC4+ visits and SBA. Additionally, attendance at least 1 family conversation during pregnancy, mode of delivery, and birth notified, significantly associated with complete CoC at the postnatal stage. Family conversation [AOR: 2.12; 95% CI: 1.56–2.88], cesarean delivery [AOR: 2.70; 95% CI: 1.82–4.02], and birth notified to HEWs [AOR: 1.95; 95% CI: 1.56–2.43] were found to be predictors of complete CoC (Table 5).

As presented above, wealth quintile, being model family, prenatal stay at MWHs, ANC booking in the first trimester, complete ANC, and administrative region were predictors for the CoC at all stages: at antepartum, intrapartum, and postpartum care. Besides, attendance at least 3 pregnancy conferences during the last pregnancy was a predictor for continued antepartum and intrapartum care. Family conversation predicted continued use of skilled delivery and postnatal care.

## Discussion

Our analysis revealed that only one-fifth of women fully utilized maternal and newborn health service packages across the continuum of care. The compliance with continuity of care across the pathway showed a significant inequality that left the poorest and those living far from health facilities behind. Moreover, uptake of complete CoC at the postpartum stage varied by mode of delivery and birth notification.

The coverage of CoC is slightly higher as compared with the findings of similar studies undertaken in Ethiopia [31, 32] and other sub-Saharan African countries [11, 14, 33]. However, it is lower than in South Asian countries [11, 19, 28]. The uptake of services declined from antepartum to postpartum stages, along with the CoC, similar to other studies [15, 17]. The CoC framework has been highlighted to link service pathways along the CoC and each contact with the health system is an opportunity to amplify the effect of subsequent contact. However, in low-income countries, service delivery is often fragmented and weakly implemented, especially during the postnatal period [4].

The importance of more frequent ANC visits for a positive pregnancy experience and its linkage with subsequent use of skilled assistance during delivery and postnatal services is well recognized [34]. Nevertheless, many women drop out from the pathway of the continuum in sub-Saharan African countries [11] including Ethiopia [31, 32]. Likewise, another significant cohort of women also dropped from skilled assistance at delivery which is one of the main strategies to ensure safe motherhood and combat maternal mortality [35, 36] indicating the country is still a long way from universal access to skilled childbirth care. The postnatal period is critical for women as they may develop life-threatening complications that can, however, be promptly treated if postnatal care is accessed [37]. There is a need to redefine the delivery strategies to strengthen the facility-based PNC, staying for at least 24-hours after delivery and providing care as recommended by WHO and the national guide [20, 38]. With the current practice of early discharge [5] and difficulties to revisit facility due to lack of transport, costs, and cultural constraints [39] women and babies would not receive appropriate CoC at the postnatal stage.

Inequity due to wealth, where the poorer women were more disadvantaged is similar to other studies conducted in low-and-middle-income countries [11, 13, 14, 17, 19, 28, 33]. That is contrary to the pro-poor strategies adopted in many countries for increasing access to services for all women to end preventable maternal and neonatal mortality [40]. Evidence also shows that when interventions are offered as a package without adequate equity considerations,

the poorest segment of the population would be very disadvantaged [41]. The regional differences in the delivery of the package of maternal and newborn health services across the continuum underlined the need to monitor CoC at the subnational level as documented in previous studies [14, 28, 33, 42].

Distance to facility influenced the odds of receiving CoC at the delivery and postpartum stages, but not for the antepartum stage. Mixed findings were reported by other studies. A study by Sakuma et al. [18] reported distance from the health facility was negatively associated with the CoC. However, a study by Mohan et al. [17] reported that distance to the nearest facility was not associated with an increased likelihood of receiving CoC. In Ethiopia, though the government expanded access to facilities and free ambulance transport, the country has poor road access, difficult terrain, and roads that are often unreachable during the rainy season. A recent study in Ethiopia affirms that women living in remote areas with no access to transportation did not have access to maternal health services [43]. This might discourage women to visit the facility for skilled birth, particularly during the night time and rainy season. Moreover, despite new health centers and primary hospitals being built in Ethiopia to achieve universal access to primary health care [21], there is still the need to further expand the reach to women for skilled delivery.

Early booking for antenatal care is the entry point for continued use of maternal and newborn health services. This is linked with the use of other maternal and newborn health services across the continuum pathway, as documented in previous studies [11, 16, 18, 28]. Early consultation and complete ANC service were also found to be associated with continued use of delivery and PNC care [10]. Uptake of complete antenatal care is connected to subsequent utilization of skilled birth attendance and postnatal care which is in line with previous studies [13]. Previous studies also documented urine samples taken were identified as predictors significantly associated with the continuation of care from having skilled birth attendants to receiving postnatal care [13, 15]. As such, program managers and policy-makers need to promote strategies and interventions to make ANC care more accessible, both in terms of early booking and content of ANC service. The level of integration and content of care at each visit should be given due attention and closely monitored.

The use of MWHs during pregnancy is found to be an independent factor regarding the continued use of skilled maternal and newborn health care in this study. Previous studies also revealed that the use of MWHs was linked to the use of maternal and newborn health services including facility birth and use of essential and emergency obstetric care [44]. This indicates a need to expand MWHs to bridge geographic barriers to access maternal health services as well as strengthen the readiness of the existing MWHs to host women postpartum and provide care in the critical period [44–46].

Being recognized as a model family was associated with the uptake of CoC. This shows the contribution of the HEP in mobilizing the community and HEWs efforts of educating families on uptake of maternal and newborn health services across the CoC pathway. Previous studies in Ethiopia also demonstrate that the model family was more likely to use maternal and newborn health services [47].

Household-level interactions with husband, neighbors, mothers-in-law, and pregnant women, as well as pregnant women's group discussion facilitated by health care workers, have positive effects on the utilization of the continuum of maternal and newborn health care. Mothers, relatives, or women development groups who informed HEWs about the birth immediately after delivery and mothers who attended at least one family conversation during pregnancy and attended at least three pregnancy conferences during the last pregnancy, were more likely to complete CoC than their counterparts. This is in agreement with a previous observational study made on collaborative community-based quality improvement

intervention in Ethiopia, which involved family meetings and labor and birth notification contributing to the increased receipt of PNC within 48 hours by skilled providers or HEWs [48]. This notification system might motivate HEWs to do home visits for the provision of PNC to the mother and the newborn.

Maternal education was significantly associated with the uptake of maternal health services at the delivery stage. However, in contrast to most studies [11, 31–33], maternal education and uptake of maternal health service across the continuum at antepartum and postpartum stages did not show significant association. While, in this study, maternal education has a positive association with the uptake of SBA [26] but with the utilization of PNC [26]. Possible explanations would be; first, educated women might book ANC late due to false self-efficacy and confidence as well as a perception that multiple visits were not necessary that might have led them to have fewer ANC visits [26]. Similarly, a study in Tanzania showed that women with primary or more education presented later in pregnancy than women with no schooling might be the reason for negative associations between education and four or more ANC visits [17]. Second, the lack of association with PNC might have related to the PNC service delivery strategy where PNC services in Ethiopia are provided mainly home-based. In line with this, some studies in Ethiopia also showed that maternal education did not have a significant association with the utilization of PNC [49, 50]. Third, we speculate that due to low-quality education, women with primary or more education in rural settings might not bring the necessary behavioral changes for better access to health or financial services than their counterparts that affects their autonomy to seek health services. Lastly, this might have also been related to the national pro-poor health policies and strategies where maternal and newborn health services in Ethiopia are provided free of charge.

The findings of this study may have been affected by social desirability and recall biases despite our efforts to minimize biases using memory aids, pretested the survey tools, trained data collectors, and allocated adequate days for data collection. Some aspects of the content of care and frequency of ANC visits might be over-reported which would positively overestimate the association between the outcome variable. In defining PNC, we included both postpartum pre-discharge care at the health facility as well as postpartum care provided after discharge to see the full spectrum of uptake of postpartum care. Accordingly, we examined predictors for pre-discharge and after-discharge care together which theoretically be different for pre-discharge care at the facility and care after discharge or at home. Another limitation of this study would be the presence of unmeasured confounders that would be correlated both with the outcome and predictor variables included in the model, particularly level-2 endogeneity arising from correlations between included individual characteristics and omitted community-level variables and/ or reverse causality of variables that would cause the model to be endogenous [51]. To minimize the potential endogeneity problems, we excluded predictor variables with the unclear direction of the effect with the outcome variables, for instance, "Complete ANC" was excluded from Model I of predicting ANC4+ uptake.

## Conclusion

The main modifiable program factors connected to the continued uptake of maternal health services include family conversation, pregnant women conference, complete ANC, ANC in the first trimester, and birth notification. As such, program managers should give focus on these practical program strategies to ensure that community or household level interactions including family conversation during pregnancy, early identification of pregnancy and link to care, and complete ANC services, notification of birth for early PNC are available to all women to improve the continuity of maternal health service use.

## Supporting information

**S1 Appendix. Survey questionnaire.** Survey questionnaire we used to collect information from study participants. The first sheet contains variable definitions (data dictionary) in English and other local languages (Amharic, Oromiffa, and Tigregna), and the second sheet contains variable answer choices.
(XLSX)

**S2 Appendix. Survey dataset.** This is survey data with variables and their values we used for the analysis.
(XLS)

## Acknowledgments

We would like to acknowledge Addis Continental Institute of Public Health (ACIPH) and University of Gondar (UoG) for providing support during the analysis and write-up. We would like to take this opportunity to acknowledge JSI Research and Training Institute Inc. /The Last Ten Kilometers Project, for allowing us to use the data for this paper. Our sincere thanks go to The Last Ten Kilometers Project of JSI Research and Training Institute Inc. staff for their contributions at all stages of implementing the research.

## Author Contributions

**Conceptualization:** Gizachew Tadele Tiruneh, Meaza Demissie, Alemayehu Worku, Yemane Berhane.

**Data curation:** Gizachew Tadele Tiruneh, Meaza Demissie, Alemayehu Worku, Yemane Berhane.

**Formal analysis:** Gizachew Tadele Tiruneh, Meaza Demissie, Alemayehu Worku, Yemane Berhane.

**Investigation:** Gizachew Tadele Tiruneh.

**Methodology:** Gizachew Tadele Tiruneh, Meaza Demissie, Alemayehu Worku, Yemane Berhane.

**Supervision:** Meaza Demissie, Alemayehu Worku, Yemane Berhane.

**Validation:** Gizachew Tadele Tiruneh, Meaza Demissie, Alemayehu Worku, Yemane Berhane.

**Visualization:** Gizachew Tadele Tiruneh, Meaza Demissie, Alemayehu Worku, Yemane Berhane.

**Writing – original draft:** Gizachew Tadele Tiruneh.

**Writing – review & editing:** Gizachew Tadele Tiruneh, Meaza Demissie, Alemayehu Worku, Yemane Berhane.

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
