## [Decision Letter · Decision Letter 0]

4 Sep 2021

PONE-D-20-37879

Predictors of maternal and newborn health service utilization across the continuum of care in Ethiopia: a multilevel analysis

PLOS ONE

Dear Dr. Tiruneh,

Thank you for submitting your manuscript to PLOS ONE. After careful consideration, we feel that it has merit but does not fully meet PLOS ONE’s publication criteria as it currently stands. Therefore, we invite you to submit a revised version of the manuscript that addresses the points raised during the review process.

Please pay particular attention to addressing the concerns and points of methodological and interpretative clarification raised by Reviewer 1.

We look forward to receiving your revised manuscript.

Kind regards,

Jamie Males

Staff Editor

PLOS ONE

Journal Requirements:

2. We note that your study is closely related to the following publication, on which you are an author:

https://bmcpregnancychildbirth.biomedcentral.com/articles/10.1186/s12884-020-03254-7

Although you have briefly cited the above study in the discussion section of your article, we feel that the scientific rationale of the current study and the contribution that it makes to the field should be better justified.  Therefore, please cite and discuss the above study in the introduction and discussion sections of your manuscript, clarifying how the present work is related to the previously published paper.

Please note that our second publication criterion states that "If a submitted study replicates or is very similar to previous work, authors must provide a sound scientific rationale for the submitted work and clearly reference and discuss the existing literature. Submissions that replicate or are derivative of existing work will likely be rejected if authors do not provide adequate justification.  http://www.plosone.org/static/publication.action#results.

Thank you for your attention to this request.

4. We noticed you have some minor occurrence of overlapping text with the following previous publication(s), which needs to be addressed:

- https://bmcpregnancychildbirth.biomedcentral.com/articles/10.1186/s12884-020-03254-7

The text that needs to be addressed involves the results section. 

In your revision ensure you cite all your sources (including your own works), and quote or rephrase any duplicated text outside the methods section. Further consideration is dependent on these concerns being addressed.

Reviewers' comments:

Reviewer's Responses to Questions

**Comments to the Author**

1. Is the manuscript technically sound, and do the data support the conclusions?

Reviewer #1: Yes

Reviewer #2: Yes

2. Has the statistical analysis been performed appropriately and rigorously? 

Reviewer #1: Yes

Reviewer #2: Yes

3. Have the authors made all data underlying the findings in their manuscript fully available?

Reviewer #1: Yes

Reviewer #2: Yes

4. Is the manuscript presented in an intelligible fashion and written in standard English?

Reviewer #1: No

Reviewer #2: Yes

5. Review Comments to the Author

Reviewer #1: I have attached a separate word document with the same comments (for correct formatting).

Review comments for PONE-D-20-37879

Overall:

This is an interesting paper and I believe the findings are important in the context of Ethiopia. However, I do see a lot of areas that need major/significant revision. I also see a lot of grammatical errors throughout the manuscript. Hence, I would suggest that the authors first address the following issues before the manuscript can be considered for publication in PLOS ONE.

Introduction:

• The following sentence needs to be revised in order to clarify what you mean

o “The continuum of maternal and newborn care provides a framework for delivering high-impact interventions organized in health service packages, ensuring appropriate linkages between family and community care, outreach and outpatient services, and the facility level across the tier to prevent maternal mortality (2).”

• CoC is not defined at first mention in the main body of the text. Please define all acronyms at first mention within the body of the text.

• I do not think that calling individual, interpersonal and household level determinants of service use “demand-side” factors is an appropriate use of the term. Instead of classifying factors as “demand-side” or “supply-side”. I would suggest just calling them what they are (i.e. individual factors, interpersonal factors, household factors, etc.)

• The author mentions that the “evidence on factors associated with adherence to the continuum of maternal and newborn health care is not widely available”. While this is largely true, there are studies done in low- and middle-income countries that looked at determinants of continuum of care. Please cite some of these studies and briefly talk about their findings to inform your study in the introduction. I see that you cite other relevant studies in the discussion section. Also, briefly mention them in the introduction section. Below is a list of references to get started:

o Determinants of continuum of care for maternal, newborn and child health services in rural Khammouane, Lao PDR by Saki Sakuma, Junko Yasuoko, Khampheng Phongluxa and Masamine Jimba (PLOS ONE; https://doi.org/10.1371/journal.pone.0215635)

o Assessing the Continuum of Care Pathway for Maternal Health in South Asia and Sub-Saharan Africa by Kavita Singh, William T. Story and Allisyn C. Moran (Maternal Child Health Journal; doi: 10.1007/s10995-015-1827-6)

o Factors associated with the continuum of care for maternal, newborn and child health in The Gambia: a cross-sectional study using Demographic and Health Survey 2013 by Jiyoung Oh, Juyoung Moon, Jae Wook Choi and Kyunghee Kim (BMJ Open; http://dx.doi.org/10.1136/bmjopen-2019-036516)

o Analysis of dropout across the continuum of maternal health care in Tanzania: findings from a cross-sectional household survey by Diwakar Mohan, Amnesty E LeFevre, Asha George, Rose Mpembeni, Eva Bazant, Neema Rusibamayila, Japhet Killewo, Peter J Winch and Abdullah H Baqui (Health Policy and Planning; https://doi.org/10.1093/heapol/czx005)

o Levels and determinants of continuum of care for maternal and newborn health in Cambodia-evidence from a population-based survey by Wenjuan Wang and Rathavuth Hong (BMC Pregnancy and Childbirth; doi: 10.1186/s12884-015-0497-0)

• The following sentence is incomplete and grammatically incorrect. Please check throughout the manuscript to correct errors like this.

o “Why the utilization of maternal and newborn health services has not improved in Ethiopia despite the huge national efforts and investments made in the last decade.”

Methods:

• For readers who may not be familiar with “Anderson’s health-seeking behavior model”, please briefly describe the model in the text.

• The author states that the outcome variable of interest was the uptake of the CoC. However, this seems like an oversimplification of what the authors actually did. I see at least three outcome variables: ANC 4+ as the outcome, continuum of care up to delivery and continuum of care through postnatal care. Please describe each outcome variable separately and in detail in the opening paragraph of the “Measurement” sub-section.

• In Table 1, the descriptions for SBA and PNC are ambiguous. For example, defining them as “proportion of women” seems incorrect as you are looking at how an individual woman responded to the corresponding questions.

• In Table 1, the author mentions that PNC included pre-discharge care at the health facility as well as care provided after discharge. Theoretically, determinants for pre-discharge care at the facility and determinants for care after discharge or at home should be treated differently. However, if you decide to keep them together, I would mention how the nature of the determinants could be different for pre-discharge care and after-discharge/home care in the discussion/limitations section.

• I would advise reconsidering the variable “Complete ANC” in the analysis models. This is because the direction of the effect between Complete ANC and ANC4+ is unclear. The authors are trying to predict ANC4+ but ANC4+ may in turn cause Complete ANC. Reverse causality causes the model to be endogenous and estimates unreliable. I would suggest omitting “Complete ANC” entirely when predicting ANC4+.

• Also, have the authors checked for multicollinearity between model family, pregnant women’s conference, family conversation and maternity waiting homes? I wonder if there is high collinearity between these variables and if there are, it could throw off standard errors. If there is high collinearity, I suggest dropping variables that are very highly collinear.

• The three sequential models are fine to use but you could also just use the entire sample for all three models:

o Model 1:

Outcome = 1 if ANC4+

Outcome = 0 if not

o Model 2:

Outcome = 1 if ANC4+ & SBA

Outcome = 0 if not

o Model 3:

Outcome = 1 if ANC4+ & SBA & PNC

Outcome = 0 if not

o If you decide to keep the three sequential models, make sure to be clear that the findings from Model 2 and Model 3 apply to a sub-sample. This means that based on your findings (from Model 2 and Model 3), program decision-makers can gain insight on a sub-population of women who have followed through to ANC (Model 2) or delivery (Model 3) but not about the general population of pregnant women. However, if you decide to use the alternative models described in the bullet points above, you can have the advantage of generalizing your findings to the general population of pregnant women.

Results:

• There are a few places where the author explains what the findings “mean”. I would suggest reserving interpretations of the findings to the discussion section. The results section should just report the findings and not try to interpret them.

• I don’t think it’s appropriate to say “CoC at the antepartum stage”. Just using antenatal care is not a “continuum of care”.

• Be clear that Model 2 and Model 3 need to be interpreted in the context of the sub-sample. For example, say “In Model 3, administrative region, wealth quintile, being model family, having booked ANC in the first trimester and having received complete ANC remained significantly associated with complete COC at the postnatal stage among women who received four or more ANC visits and skilled birth attendance.”

Discussion:

• Overall, the discussion section is well-organized, rich and clearly written.

• If you decide to keep the three sequential models, I would ask the authors make it very clear in the discussion section that the findings (particularly from Model 2 and Model 3) are not generalizable to the larger population. Rather, those findings (from Model 2 and Model 3) specifically apply to a sub-sample of women who have followed through to ANC (Model 2) or delivery (Model 3). Hence, programmatic implications are only applicable to those women who have followed through to ANC or delivery and not to the general population of pregnant women.

• I would also focus on fixing grammatical errors and awkward sentences.

• Lastly, I would suggest expanding the discussion of methodological limitations a little bit (i.e. potential issues of endogeneity).

Reviewer #2: the authors investigated the predictors of maternal and newborn health service utilization across the continuum of care in Ethiopia. the authors did a great work and have added to the body of knowledge on maternal health.

the authors should edit their writeup and make sure where ever they use an abbreviation for the first time, they first indicate it in full and the abbreviation in bracket.

6. PLOS authors have the option to publish the peer review history of their article (what does this mean?). If published, this will include your full peer review and any attached files.

Reviewer #1: No

Reviewer #2: No

---

## [Author Response · Author response to Decision Letter 0]

8 Nov 2021

A point-by-point response to reviewers' comments

Journal: PLOS ONE

PONE-D-20-37879

Predictors of maternal and newborn health service utilization across the continuum of care in Ethiopia: a multilevel analysis

PLOS ONE

The authors would like to appreciate and thank the reviewers and editors for the constructive comments. Our point-by-point responses are below each of the comments in italics. 

Reviewer #1: 

Overall:

This is an interesting paper and I believe the findings are important in the context of Ethiopia. However, I do see a lot of areas that need major/significant revision. I also see a lot of grammatical errors throughout the manuscript. Hence, I would suggest that the authors first address the following issues before the manuscript can be considered for publication in PLOS ONE.

Introduction:

• The following sentence needs to be revised in order to clarify what you mean

o “The continuum of maternal and newborn care provides a framework for delivering high-impact interventions organized in health service packages, ensuring appropriate linkages between family and community care, outreach and outpatient services, and the facility level across the tier to prevent maternal mortality (2).”

Thanks for the comment. To clarify for the readers, it is revised as follows, “The continuum of maternal and newborn care is highlighted as a framework for delivering high-impact interventions across the pregnancy, childbirth, and postpartum life-course and the level of service delivery to prevent maternal and newborn mortalities (2).”

• CoC is not defined at first mention in the main body of the text. Please define all acronyms at first mention within the body of the text.

Comment well taken and addressed in this version.

• I do not think that calling individual, interpersonal and household level determinants of service use “demand-side” factors is an appropriate use of the term. Instead of classifying factors as “demand-side” or “supply-side”. I would suggest just calling them what they are (i.e. individual factors, interpersonal factors, household factors, etc.)

Many thanks for the comment. Comment well taken and addressed in this version. 

• The author mentions that the “evidence on factors associated with adherence to the continuum of maternal and newborn health care is not widely available”. While this is largely true, there are studies done in low- and middle-income countries that looked at determinants of continuum of care. Please cite some of these studies and briefly talk about their findings to inform your study in the introduction. I see that you cite other relevant studies in the discussion section. Also, briefly mention them in the introduction section. Below is a list of references to get started:

o Determinants of continuum of care for maternal, newborn and child health services in rural Khammouane, Lao PDR by Saki Sakuma, Junko Yasuoko, Khampheng Phongluxa and Masamine Jimba (PLOS ONE; https://doi.org/10.1371/journal.pone.0215635)

o Assessing the Continuum of Care Pathway for Maternal Health in South Asia and Sub-Saharan Africa by Kavita Singh, William T. Story and Allisyn C. Moran (Maternal Child Health Journal; doi: 10.1007/s10995-015-1827-6)

o Factors associated with the continuum of care for maternal, newborn and child health in The Gambia: a cross-sectional study using Demographic and Health Survey 2013 by Jiyoung Oh, Juyoung Moon, Jae Wook Choi and Kyunghee Kim (BMJ Open; http://dx.doi.org/10.1136/bmjopen-2019-036516)

o Analysis of dropout across the continuum of maternal health care in Tanzania: findings from a cross-sectional household survey by Diwakar Mohan, Amnesty E LeFevre, Asha George, Rose Mpembeni, Eva Bazant, Neema Rusibamayila, Japhet Killewo, Peter J Winch and Abdullah H Baqui (Health Policy and Planning; https://doi.org/10.1093/heapol/czx005)

o Levels and determinants of continuum of care for maternal and newborn health in Cambodia-evidence from a population-based survey by Wenjuan Wang and Rathavuth Hong (BMC Pregnancy and Childbirth; doi: 10.1186/s12884-015-0497-0)

Many thanks for the valid comments. All these articles are cited in the Background and Discussion sections in the revised version.

• The following sentence is incomplete and grammatically incorrect. Please check throughout the manuscript to correct errors like this.

o “Why the utilization of maternal and newborn health services has not improved in Ethiopia despite the huge national efforts and investments made in the last decade.”

Comment well taken and addressed. 

Methods:

• For readers who may not be familiar with “Anderson’s health-seeking behavior model”, please briefly describe the model in the text.

Comment is well taken and a brief description of the model included. 

• The author states that the outcome variable of interest was the uptake of the CoC. However, this seems like an oversimplification of what the authors actually did. I see at least three outcome variables: ANC 4+ as the outcome, continuum of care up to delivery and continuum of care through postnatal care. Please describe each outcome variable separately and in detail in the opening paragraph of the “Measurement” sub-section.

Comment well taken and addressed as follows, “The outcome variables of interest of the study were the uptake of the CoC at antepartum, intrapartum, and postpartum stages: 1) continuum of care at the antepartum stage is women who received ANC 4+ visits, 2) continuum of care at the intrapartum stage is those women who continued use of skilled birth attendance after receiving ANC4+ visits, and 3) continuum of care at the postpartum stage or complete continuum is those women who received PNC for the mothers and their newborns, within six weeks of their delivery (either in a facility or at home) after receiving both ANC4+ visits and delivered by a skilled assistance.”

• In Table 1, the descriptions for SBA and PNC are ambiguous. For example, defining them as “proportion of women” seems incorrect as you are looking at how an individual woman responded to the corresponding questions.

Comment is well taken and addressed in this version.

• In Table 1, the author mentions that PNC included pre-discharge care at the health facility as well as care provided after discharge. Theoretically, determinants for pre-discharge care at the facility and determinants for care after discharge or at home should be treated differently. However, if you decide to keep them together, I would mention how the nature of the determinants could be different for pre-discharge care and after-discharge/home care in the discussion/limitations section.

Thanks for the valuable comment. In the discussion section, we included the following sentence as a limitation of this paper. “In defining PNC, we included both postpartum pre-discharge care at the health facility as well as postpartum care provided after discharge to see the full spectrum of uptake of postpartum care. Accordingly, we examined predictors for pre-discharge and after-discharge care together which theoretically be different for pre-discharge care at the facility and care after discharge or at home.”

• I would advise reconsidering the variable “Complete ANC” in the analysis models. This is because the direction of the effect between Complete ANC and ANC4+ is unclear. The authors are trying to predict ANC4+ but ANC4+ may in turn cause Complete ANC. Reverse causality causes the model to be endogenous and estimates unreliable. I would suggest omitting “Complete ANC” entirely when predicting ANC4+.

Thanks for the valid and educational comment. It is well taken and Complete ANC is omitted from Model I. The results including the result Table are edited accordingly. 

• Also, have the authors checked for multicollinearity between model family, pregnant women’s conference, family conversation and maternity waiting homes? I wonder if there is high collinearity between these variables and if there are, it could throw off standard errors. If there is high collinearity, I suggest dropping variables that are very highly collinear.

Thanks a lot for the valid comments. We have checked for multi-collinearity to see the assumption of independence of variables in the multiple regression model and we found out a mean-variance inflation factor (vif) of 1.05 that there is no multi-collinearity problem. 

• The three sequential models are fine to use but you could also just use the entire sample for all three models:

o Model 1:

Outcome = 1 if ANC4+

Outcome = 0 if not

o Model 2:

Outcome = 1 if ANC4+ & SBA

Outcome = 0 if not

o Model 3:

Outcome = 1 if ANC4+ & SBA & PNC

Outcome = 0 if not

o If you decide to keep the three sequential models, make sure to be clear that the findings from Model 2 and Model 3 apply to a sub-sample. This means that based on your findings (from Model 2 and Model 3), program decision-makers can gain insight on a sub-population of women who have followed through to ANC (Model 2) or delivery (Model 3) but not about the general population of pregnant women. However, if you decide to use the alternative models described in the bullet points above, you can have the advantage of generalizing your findings to the general population of pregnant women.

Thanks for the detailed explanation. We defined the three models exactly how you described them above. Few differences in the sample size for the three models are due to missing values where we recoded “Don’t know” values as missing. In the analysis section, we clearly defined how the outcome variables are coded 1 or 0 for further clarification.

Results:

• There are a few places where the author explains what the findings “mean”. I would suggest reserving interpretations of the findings to the discussion section. The results section should just report the findings and not try to interpret them.

Comment is well taken and revisited in this version. 

• I don’t think it’s appropriate to say “CoC at the antepartum stage”. Just using antenatal care is not a “continuum of care”.

Comment is well taken and revisited in this version. 

• Be clear that Model 2 and Model 3 need to be interpreted in the context of the sub-sample. For example, say “In Model 3, administrative region, wealth quintile, being model family, having booked ANC in the first trimester and having received complete ANC remained significantly associated with complete COC at the postnatal stage among women who received four or more ANC visits and skilled birth attendance.”

Comment well taken and revisited in this version. 

Discussion:

• Overall, the discussion section is well-organized, rich and clearly written.

Thanks much.

• If you decide to keep the three sequential models, I would ask the authors make it very clear in the discussion section that the findings (particularly from Model 2 and Model 3) are not generalizable to the larger population. Rather, those findings (from Model 2 and Model 3) specifically apply to a sub-sample of women who have followed through to ANC (Model 2) or delivery (Model 3). Hence, programmatic implications are only applicable to those women who have followed through to ANC or delivery and not to the general population of pregnant women.

Thanks for the comment. As we explained earlier we defined the three models exactly how you described them above. In this version, we clearly defined how the outcome variables are coded 1 or 0. 

• I would also focus on fixing grammatical errors and awkward sentences.

Comment well taken and addressed. 

• Lastly, I would suggest expanding the discussion of methodological limitations a little bit (i.e. potential issues of endogeneity).

Thanks for the valid comment. The following sentences are included in the limitation section of the revised manuscript. “Another limitation of this study would be the presence of unmeasured confounders that would be correlated both with the outcome and predictor variables included in the model, particularly level-2 endogeneity arising from correlations between included individual characteristics and omitted community-level variables and/ or reverse causality of variables that would cause the model to be endogenous [52]. To minimize the potential endogeneity problems, we excluded predictor variables with the unclear direction of the effect with the outcome variables, for instance, “Complete ANC” was excluded from Model I of predicting ANC4+ uptake.” Line 401-8; page 27

Reviewer #2: the authors investigated the predictors of maternal and newborn health service utilization across the continuum of care in Ethiopia. the authors did a great work and have added to the body of knowledge on maternal health.

Thanks!!

the authors should edit their writeup and make sure where ever they use an abbreviation for the first time, they first indicate it in full and the abbreviation in bracket.

Comment is well taken and addressed in this version.

---

## [Decision Letter · Decision Letter 1]

6 Dec 2021

PONE-D-20-37879R1Predictors of maternal and newborn health service utilization across the continuum of care in Ethiopia: a multilevel analysisPLOS ONE

Dear Dr. Tiruneh,

Thank you for submitting your manuscript to PLOS ONE. After careful consideration, we feel that it has merit but does not fully meet PLOS ONE’s publication criteria as it currently stands. Therefore, we invite you to submit a revised version of the manuscript that addresses the points raised during the review process.

We look forward to receiving your revised manuscript.

Kind regards,

Orvalho Augusto, MD, MPH

Academic Editor

PLOS ONE

Journal Requirements:

Additional Editor Comments (if provided):

This is an interesting work. They choose 3 indicators and analyse them as a cascade of a continuum of care (CoC) antepartum (4 antenatal visits), intrapartum (skilled birth attendance conditional the woman had the 4 antenatal visits), and postpartum (postnatal care among those who completed the intrapartum step as defined here). Then they present prevalence of these indicators per characteristics and elements chosen from Anderson’s health-seeking behaviour model.

However, few issues:

1. Please add some description of the administrative division of Ethiopia. At least the reader will understand what is a woreda.

2. Please provide 95% confidence intervals (CI) for each step. That could be accomplished with a bar plot for the antepartum, intrapartum and postpartum. Please, make sure that those CI are adjusted for the clustering.

a. Tables 3 and 4 could have those 95% confidence intervals rather than the p-values. Again, make sure those CI are adjusted for the cluster sampling.

3. Table 3 - for education, I believe complete continuum figures are a repetition of the “continued at antenatal care”.

4. Table 5 - Please add a row with the number of observations included in each model

Reviewers' comments:

Reviewer's Responses to Questions

**Comments to the Author**

1. If the authors have adequately addressed your comments raised in a previous round of review and you feel that this manuscript is now acceptable for publication, you may indicate that here to bypass the “Comments to the Author” section, enter your conflict of interest statement in the “Confidential to Editor” section, and submit your "Accept" recommendation.

Reviewer #3: (No Response)

2. Is the manuscript technically sound, and do the data support the conclusions?

Reviewer #3: Partly

3. Has the statistical analysis been performed appropriately and rigorously? 

Reviewer #3: Yes

4. Have the authors made all data underlying the findings in their manuscript fully available?

Reviewer #3: No

5. Is the manuscript presented in an intelligible fashion and written in standard English?

Reviewer #3: No

6. Review Comments to the Author

Reviewer #3: As stated by a previous reviewer, the entire manuscript needs copy editing. Specific issues include:

Line 47-48: This first sentence is confusing and frankly, not necessary.

Line 50-51: I believe a modifier may be missing here, such as “level of service delivery needed to prevent…”

Line 63: What are “supply-side” factors? Number of physicians/nurses/facilities?

Line 68: Needs a reference

Line 124-128: The direction of “no difficulty of distance and transportation arrangement to access medical care” as an “exposure” variable does not make sense given the direction of the other variables; please explain. Additionally, II am not sure that “exposure” is the correct term. Predictor is more appropriate. You were not “exposing” women to these factors with an intervention.

Line 134-136: At line 135, “four or more antental care (ANC4+)” appears in item 2 but should appear in item 1, as that is its first use.

Line 137: What is PNC? (Spell it out before abbreviating it.)

Line 149: I am surprised you did not include the infant’s birth weight, estimated gestational age at birth, health at birth (did the infant need to stay in a neonatal special care unit, did the infant have a congenital anomaly), or survival. These factors may influence receipt of care at the delivery and postnatal stages. Were these items asked?

Table 3, 4: What are these proportions? Neither columns nor rows sum to 100% in any group. Also, what is the denominator for the N’s?

7. PLOS authors have the option to publish the peer review history of their article (what does this mean?). If published, this will include your full peer review and any attached files.

Reviewer #3: No

---

## [Author Response · Author response to Decision Letter 1]

24 Jan 2022

A point-by-point response to reviewer/editorial

Journal: PLOS ONE

PONE-D-20-37879R1

Predictors of maternal and newborn health service utilization across the continuum of care in Ethiopia: a multilevel analysis

The authors would like to appreciate and thank the reviewers and editors for the constructive comments. Our point-by-point responses are below each of the comments in italics. 

Journal Requirements:

Thank you for the valid comments. References cited are now edited. Articles under review

Additional Editor Comments (if provided):

This is an interesting work. They choose 3 indicators and analyse them as a cascade of a continuum of care (CoC) antepartum (4 antenatal visits), intrapartum (skilled birth attendance conditional the woman had the 4 antenatal visits), and postpartum (postnatal care among those who completed the intrapartum step as defined here). Then they present prevalence of these indicators per characteristics and elements chosen from Anderson’s health-seeking behaviour model.

Thank you so much.

However, few issues:

1. Please add some description of the administrative division of Ethiopia. At least the reader will understand what is a woreda.

Comment well taken. A paragraph is added under the Methods section to describe the administrative division of the country. (Line 79-81, page 5)

2. Please provide 95% confidence intervals (CI) for each step. That could be accomplished with a bar plot for the antepartum, intrapartum and postpartum. Please, make sure that those CI are adjusted for the clustering.

Comment well taken. The continuum of care coverage across antepartum, intrapartum and postpartum with error bars is presented in Figure 1, separately attached. 

a. Tables 3 and 4 could have those 95% confidence intervals rather than the p-values. Again, make sure those CI are adjusted for the cluster sampling.

Thank you for the valid comments. The 95% CI adjusted for the cluster sampling is included in both Tables 3 and 4.

3. Table 3 - for education, I believe complete continuum figures are a repetition of the “continued at antenatal care”.

Thanks a lot for comment. You are right. It was an error. It is now fixed.

4. Table 5 - Please add a row with the number of observations included in each model

Thanks. Number of observations in each model is included in Table 5.

Comments to the Author

Reviewer #3: As stated by a previous reviewer, the entire manuscript needs copy editing. Specific issues include:

Comment well taken and copy edited. 

Line 47-48: This first sentence is confusing and frankly, not necessary.

Comment well taken and edited accordingly. 

Line 50-51: I believe a modifier may be missing here, such as “level of service delivery needed to prevent…”

Comment well taken and edited accordingly. 

Line 63: What are “supply-side” factors? Number of physicians/nurses/facilities?

Comment well taken edited accordingly. 

Line 68: Needs a reference

Comment well taken.

Line 124-128: The direction of “no difficulty of distance and transportation arrangement to access medical care” as an “exposure” variable does not make sense given the direction of the other variables; please explain. Additionally, II am not sure that “exposure” is the correct term. Predictor is more appropriate. You were not “exposing” women to these factors with an intervention.

Thanks for the comment. We checked the adequacy of our sample size using a double population, using case and control or exposure and non-exposed groups to determine the predictor variables of the study. 

Line 134-136: At line 135, “four or more antental care (ANC4+)” appears in item 2 but should appear in item 1, as that is its first use.

Comment well taken and addressed.

Line 137: What is PNC? (Spell it out before abbreviating it.)

Comment well taken and spelt out.

Line 149: I am surprised you did not include the infant’s birth weight, estimated gestational age at birth, health at birth (did the infant need to stay in a neonatal special care unit, did the infant have a congenital anomaly), or survival. These factors may influence receipt of care at the delivery and postnatal stages. Were these items asked?

Thanks for the valid comments. Variable infant’s birth weight was collected and included in the analysis; but not significantly associated with COC at postpartum stage. And this is indicated in the revised version of the manuscript. However, we did not collect variables like estimated gestational age at birth, health at birth and/or presence of a congenital anomaly.

Table 3, 4: What are these proportions? Neither columns nor rows sum to 100% in any group. Also, what is the denominator for the N’s?

Thank you for the comment. The denominator for Tables 3 and 4 are presented as “Total’’ column at the end of each table. The proportions presented in these tables are actual percentages of each category; their complements are not presented which we believe readers can get by subtracting from 100%.

---

## [Decision Letter · Decision Letter 2]

15 Feb 2022

Predictors of maternal and newborn health service utilization across the continuum of care in Ethiopia: a multilevel analysis

PONE-D-20-37879R2

Dear Dr. Tiruneh,

We’re pleased to inform you that your manuscript has been judged scientifically suitable for publication and will be formally accepted for publication once it meets all outstanding technical requirements.

Kind regards,

Orvalho Augusto, MD, MPH

Academic Editor

PLOS ONE

Additional Editor Comments (optional):

Reviewers' comments:

Reviewer's Responses to Questions

**Comments to the Author**

1. If the authors have adequately addressed your comments raised in a previous round of review and you feel that this manuscript is now acceptable for publication, you may indicate that here to bypass the “Comments to the Author” section, enter your conflict of interest statement in the “Confidential to Editor” section, and submit your "Accept" recommendation.

Reviewer #3: (No Response)

2. Is the manuscript technically sound, and do the data support the conclusions?

Reviewer #3: Yes

3. Has the statistical analysis been performed appropriately and rigorously? 

Reviewer #3: Yes

4. Have the authors made all data underlying the findings in their manuscript fully available?

Reviewer #3: Yes

5. Is the manuscript presented in an intelligible fashion and written in standard English?

Reviewer #3: Yes

6. Review Comments to the Author

Reviewer #3: I was not clear about my previous concern about lines 126-127.

The "exposure" variables that you list indicate problems - lack of autonomy, no media exposure, no maternal education, poorest wealth quintile. You also include "no difficulty of distance to access medical care" and "no difficulty of transport arrangement to access medical care." I believe that, given the direction of the other variables, that you mean "difficulty of distance and transport arrangement to access medical care".

If I am correct, at lines 128 and 129, you will want to remove "having difficulty of distance and transport arrangement to access medical care" and replace it with "no difficulty of distance or transport arrangement to access medical care."

Otherwise, thank you for the excellent revision.

7. PLOS authors have the option to publish the peer review history of their article (what does this mean?). If published, this will include your full peer review and any attached files.

Reviewer #3: No

---

## [Editor Report · Acceptance letter]

17 Feb 2022

PONE-D-20-37879R2 

Predictors of maternal and newborn health service utilization across the continuum of care in Ethiopia: a multilevel analysis 

Dear Dr. Tiruneh:

I'm pleased to inform you that your manuscript has been deemed suitable for publication in PLOS ONE. Congratulations! Your manuscript is now with our production department. 

Kind regards, 

on behalf of

Dr. Orvalho Augusto 

Academic Editor

PLOS ONE